 

# Trait-based meta-analysis of microbial guilds in the iron redox cycle

**Fernando Díaz-González,**[1,2] **Camila Rojas-Villalobos,**[1,3] **Francisco Issotta,**[1,4] **Sofía Reyes-Impellizzeri,**[1,2] **Sabrina Hedrich,**[5] **D. Barrie Johnson,**[6,7,8] **Pedro Temporetti,**[9] **Raquel Quatrini**[1,10]

**ABSTRACT**   Microbial iron (Fe) redox cycling underpins key biogeochemical processes, yet the functional diversity, ecological roles, and trait architectures of iron-transforming microbes remain poorly synthesized across global environments. Here, we present a systematic review and trait-based meta-analysis of 387 microbial taxa spanning 314 studies and 76 years of research, integrating phenotypic, genomic, and environmental data to define ecologically coherent microbial iron redox cycle guilds. Rather than relying on taxonomy, our framework delineates first-order functional guilds— Fe(III) reducers, Fe(II) oxidizers, and dual-capacity Fe oxidizers/reducers—and resolves second-order guilds based on trait syndromes, such as acidophily, redox flexibility, or metabolic breadth. Trait profiling revealed that iron-cycling capacities frequently transcend phylogenetic boundaries, with multiple guilds converging in chemically stratified hotspots like hot springs, hydrothermal vents, and acid mine drainages. Dual-capacity Fe oxidizers/reducers (e.g., *Acidithiobacillus ferrooxidans* and *Metallosphaera sedula*) emerged as overlooked mediators of "cryptic" iron cycling, possessing genomic repertoires capable of toggling between oxidative and reductive modes in response to redox oscillations. Hierarchical clustering and kernel density analyses of ecophysiological traits highlighted niche partitioning along key environmental filters, including pH, iron availability, salinity, and temperature. Collectively, this work introduces the Guild Exploitation Pattern as a conceptual lens for understanding iron microbiome assembly, providing a data-driven foundation for predicting microbial contributions to iron cycling under changing environmental conditions.

**IMPORTANCE**   Iron redox reactions shape nutrient turnover, contaminant mobility, and primary productivity, yet the microbes driving these processes are often studied in isolation. By integrating decades of data into a trait-based guild framework, we reveal the ecophysiological diversity and niche differentiation of microbial iron redox cycling taxa across environments. Our synthesis exposes major gaps, such as limited trait data for >80% of dual-capacity Fe oxidizing/reducing species and highlights the need for functional trait surveys to complement metagenomics and cultivation efforts. The guild framework presented here advances predictive microbial ecology by linking metabolic traits with environmental gradients, offering a robust foundation for incorporating iron cycling into ecosystem models and biogeochemical forecasts.

**KEYWORDS**   microbial iron redox cycle, trait-based ecology, guild exploitation pattern, meta-analysis, biogeochemical cycling

Iron (Fe) is a critical redox-active element in Earth's biogeochemical cycles (1), shaping nutrient turnover (2) and microbial ecosystem functioning (3–5). Its cycling is tightly linked to environmental chemistry and microbial metabolism, with important implications for ecosystem processes and resilience under environmental change. For

**Peer Reviewers** Ricardo Amils, Universidad Autonoma de Madrid Centro de Biologia Molecular Severo Ochoa, Madrid, Spain; Michael Christopher Macey, The Open University, Milton Keynes, United Kingdom

Address correspondence to Raquel Quatrini, rquatrini@cienciavida.org, or Pedro Temporetti, temporettipf@comahue-conicet.gob.ar.

The authors declare no conflict of interest.

See the funding table on p. 20.

instance, atmospheric fluxes of soluble Fe from anthropogenic sources influence marine biogeochemistry and productivity (6–8). Fe occurs in three oxidation states: Fe(0), Fe(II), and Fe(III). Among these species, Fe(II) and Fe(III) predominate in nature and occur as aqueous ions, organically complexed forms, or as components of diverse mineral phases. These forms differ in solubility and bioavailability, depending on pH, oxygen levels, and the presence of ligands or mineral surfaces (9–11).

Microorganisms actively participate in the interconversion between Fe(II) and Fe(III), a process often referred to as the "iron redox wheel" (1, 12). Their metabolic activity can greatly accelerate these redox reactions; e.g., microbial Fe(II) oxidation under acidic conditions can proceed orders of magnitude faster than abiotic processes (13). Microbes capable of coupling their energy metabolism to iron transformations span both the archaeal and bacterial domains and thrive under a wide range of physicochemical conditions (14). These taxa are collectively referred to as members of the microbial iron redox cycle (MIRC), encompassing both Fe(III) reducers (RED) and Fe(II) oxidizers (OXI), along with microorganisms that exploit iron as an electron donor or acceptor, depending on environmental context (dual-capacity Fe oxidizers/reducers [O/R]). Since their initial recognition in the 1970s, a large and steadily growing number of MIRC taxa have been formally described in the literature, at varying levels of physiological, ecological, and genomic detail, highlighting the breadth of microbial adaptations and strategies involved in Fe redox cycling.

At circumneutral pH, dissimilatory Fe(III) reduction is widespread among anaerobes such as *Geobacter* spp. (15, 16) and facultative aerobes like *Shewanella*, which can use extracellular electron transfer to respire Fe(III) in sediments or biofilm-associated microniches (17, 18). Microaerophilic Fe(II) oxidizers, such as *Gallionella* or *Leptothrix*, dominate iron-rich groundwater and microoxic zones of sediments at neutral pH, while in stratified aquatic environments, phototrophic Fe(II) oxidizers, such as *Chlorobium ferrooxidans* or *Rhodopseudomonas palustris*, exploit Fe(II) under light-driven anaerobic conditions. Other Fe(II)-oxidizing bacteria operate under anoxic, nitrate-reducing conditions (*Thiobacillus denitrificans* and *Acidovorax* spp.), particularly in subsurface aquifers (3). In acidic environments, where Fe(II) remains stable and soluble under aerobic conditions, acidophilic Fe(II)-oxidizing *Acidithiobacillus*, *Leptospirillum*, or *Ferrovum* spp. abound (19, 20). Acidophilic Fe(III) reducers, such as *Acidiphilium*, *Acidocella*, and *Acidobacterium* spp., oxidize organic compounds at low pH, contributing to complete Fe cycling in low-oxygen, acidic microenvironments (21–23).

These diverse iron redox strategies enable microorganisms to occupy niches defined by contrasting oxygen, carbon, and iron availability. In such settings, redox transformations can proceed simultaneously across fine-scale spatial microgradients, giving rise to "cryptic" Fe cycling (3, 24–26). This redox heterogeneity imposes strong environmental filtering, favoring distinct trait combinations and the functional structure of iron-cycling microbial communities. Ecological studies of the iron cycle indicate that iron-transforming microbes often co-occur in spatially structured habitats, such as iron mats, sediments, and redox-stratified biofilms, where they engage in both competitive and cooperative interactions that shape local biogeochemical dynamics (27–30). Fe(II)-oxidizing and Fe(III)-reducing taxa frequently occupy complementary niches, exploiting microscale oxygen or iron gradients, while cross-feeding and shared public goods (e.g., siderophores) facilitate guild-level interactions (e.g., between iron, sulfur, nitrogen, and carbon cycling microbes). Despite the growing recognition of these complex ecological dynamics, the concept of microbial guilds, i.e., a functionally coherent group of organisms that exploit, produce, or transform a key resource under shared environmental constraints, regardless of their phylogenetic affiliation, ecological origin, or specific functional mechanisms (31, 32), remains underutilized and largely implicit in Fe microbiome research (e.g. see reference 33), underscoring the need for a trait-based framework to capture functional patterns and adaptive strategies in iron redox communities.

In this study, we aimed to assess current public knowledge regarding the MIRC guilds through a trait-based lens via a systematic meta-analysis of species-level literature and cognate genomes. This included data mining, wrangling, curation, analysis, and visualization to observe and unravel competing and co-occurring taxa, frame their niches, and identify patterns of resource use and environmental adaptation. By doing so, we aimed to provide a first approximation of the MIRC Guild Exploitation Pattern, i.e., the multidimensional trait space encompassing the environmental tolerances and metabolic capabilities that define the functional niche of iron redox guild members. This is essential to elucidate the mechanisms of persistence, assembly, and functional contributions to biogeochemical cycling across environmental contexts or in the face of changing conditions.

## MATERIALS AND METHODS

### Systematic literature search on MIRC guilds

Studies related to the *in vitro* characterization and description of strains related to the MIRC were included as eligibility criteria, without restriction of the publication date. As sources of information, the Web of Science, SCOPUS, PubMed, and PubMed Central databases were reviewed. Also, manual and cross-reference searches were carried out with the Google search engine using the double quotes operator (" ") to refine the search and the operator (*site:*) to perform specific searches on web pages of interest, such as ncbi.nlm.nih.gov and microbiologyresearch.org. The search criteria, or keywords, were Archaea, bacteria, Fe(II), Fe(II)-oxidizing, Fe(III), Fe(III)-reducing, $Fe^{2+}$, $Fe^{3+}$, ferric, ferrous, iron, iron-oxidizing, iron-reducing, iron(II), iron(II)-oxidizing, iron(III), iron(III)-reducing, oxidation, oxidizing, reducing, reduction, review and sp. nov. To complement the literature-derived information, data were retrieved from the BacDive (34) and MediaDive (35) databases, which integrate comprehensive taxonomic, ecological, physiological, molecular, and applied metadata for cultured microorganisms. Article screening and data recruitment were completed on 24 January 2025. Data mining and tabulation were performed mostly manually. For strains cataloged in MediaDive, total iron concentrations were obtained with a semiautomatic pipeline using a custom R script available via FigShare (doi:10.6084/m9.figshare.30251776). These values correspond to the total Fe supplied in culture media formulations during physiological characterization and do not represent experimentally determined physiological optima, which are rarely quantified explicitly in strain descriptions. Accordingly, these values are interpreted as standardized proxies of cultivation context rather than true optima for iron utilization.

### Categorization of environmental and lifestyle traits of MIRC guild members

For each described species, habitat and source material categories were manually assigned by classifying sampling site descriptions using the ENVO ontology (36). To assign categories related to microbial lifestyles, a set of descriptors was constructed based on the optimal growth range for each biophysicochemical variable through a literature analysis. These phenotypic descriptors cover the definitions of pH, temperature (°C), salinity (% wt/vol NaCl), oxygen availability, trophism, and redox activity. Furthermore, the remaining assigned categories were defined using abbreviations derived from the author's descriptions (Table 1).

### Profiling of functional genomic traits

Public genomes (*n* = 293) were obtained from the public NCBI Genome database (https://www.ncbi.nlm.nih.gov/datasets/genome/) on 29 May 2025 (Table S1), using the "data sets" command-line tool (37), with default parameters and genome accession as query. Fe-cycling genes were identified from genome assemblies using FeGenie (v.1.2) (38), with default parameters and the normalization option for the output table.

**TABLE 1** Definitions of microbial lifestyles based on optimal growth range per ecophysiological and physicochemical variables

| Variable | Optimal growth range (non-optimal) or description | Lifestyle | Abbreviation |
|---|---|---|---|
| pH | ≤1 | Hyperacidophilic | HACI |
| | >1 and ≤3 | Extremely acidophilic | EACI |
| | >3 and ≤5 | Moderately acidophilic | MACI |
| | ≤5 | Acidophilic | ACI |
| | >5 (≤5) | Acid tolerant | ACIT |
| | >5 and <9 | Neutrophilic | NEU |
| | <9 (≥9) | Alkali tolerant | ALKT |
| | ≥9 | Alkaliphilic | ALK |
| | ≥9 and <11 | Moderately alkaliphilic | MALK |
| | ≥11 and <13 | Extremely alkaliphilic | EALK |
| | ≥13 | Hyperalkaliphilic | HALK |
| Temperature (°C) | ≤−20 | Hyperpsychrophilic | HPSY |
| | >−20 and ≤0 | Extremely psychrophilic | EPSY |
| | >0 and ≤20 | Moderately psychrophilic | MPSY |
| | ≤20 | Psychrophilic | PSY |
| | >20 (≤20) | Psychrotolerant | PSYT |
| | >20 and <40 | Mesophilic | MES |
| | <40 (≥40) | Thermotolerant | THET |
| | ≥40 | Thermophilic | THE |
| | ≥40 and <60 | Moderately thermophilic | MTHE |
| | ≥60 and <80 | Extremely thermophilic | ETHE |
| | ≥80 | Hyperthermophilic | HTHE |
| Salinity (% wt/vol NaCl) | <1 (≤1) | Non-halo tolerant | NHAT |
| | <1 (>1 and ≤15) | Halo tolerant | HAT |
| | ≥1 | Halophilic | HAL |
| | ≥1 and <3 | Slightly halophilic | SHAL |
| | ≥3 and <15 | Moderately halophilic | MHAL |
| | ≥15 and <30 | Extremely halophilic | EHAL |
| | ≥30 | Hyperhalophilic | HHAL |
| Oxygen availability | Grow only in the absence of oxygen | Anaerobic | ANB |
| | Can grow aerobically or anaerobically | Facultative | FAC |
| | Grow at low levels of oxygen, not in air | Microaerobic | MIA |
| | Grow in ≥21% oxygen, level present in air | Aerobic | AEB |
| Trophism | Energy is chiefly provided by photochemical reaction. | Phototrophic | PHO |
| | Energy is provided entirely by dark chemical reaction. | Chemotrophic | CHE |
| | Electrons are provided by exogenous inorganic substances. | Lithotrophic | LIT |
| | Electrons are provided by exogenous organic substances. | Organotrophic | ORG |
| | All essential metabolites are synthesized. | Autotrophic | AUT |
| | Not all essential metabolites are synthesized. | Heterotrophic | HET |
| | Energy and/or carbon are derived from both inorganic and organic sources. | Mixotrophic | MIX |
| Redox activity | Oxidation of iron | Fe(II)-oxidizer | OXI |
| | Reduction of iron | Fe(III)-reducer | RED |
| | Oxidation and reduction | Dual-capacity Fe-oxidizer/reducer | O/R |

## Phylogenetic reconstruction

The 16S rRNA gene sequences used in phylogenetic inference of the MIRC guilds are listed in Table S1. Sequences were recovered from the nucleotide database in NCBI (https://www.ncbi.nlm.nih.gov/nuccore/) or retrieved from the recovered genomes. A minority of sequences ($n = 2$, sp-234 and sp-237) were also predicted on 22 May 2025 using the RAST server with the RASTtk annotation scheme with default settings (39). Small subunit ribosomal RNA gene sequences were aligned using MAFFT (v.7.310) software with the FFT-NS-2 method (40), with default parameters and the reorder aligned option. Phylogenetic trees were inferred using maximum likelihood (ML) as implemented in MEGA (v.11.0.13) (41), applying the Tamura–Nei model and the nearest-neighbor interchange (NNI) heuristic for ML tree optimization under default settings.The resulting ML tree was rooted with the Archaea domain branch and depicted as a cladogram. The supporting alignment and the original, unaltered cladogram, preserving full evolutionary distances and branch length information, are available in Newick format via FigShare (42).

## Trait data processing, statistical analysis, and visualization

Data analysis and visualization were conducted using the R language (R v.4.4.2) through RStudio (v.2024.12.1.563) (43). Devtools (v.2.4.5) was used for development, testing, and installation of R packages (44). Data wrangling was performed with fastDummies (v.1.7.5) (45); janitor (v.2.2.1) was used in cleaning and standardization of variable names and data tables (46); readr (v.2.1.5) allowed fast and efficient import of text files and tabular data into R (47); readxl (v.1.4.5) was used for direct reading of Excel files (48); and tidyverse (v.2.0.0) allowed data manipulation, visualization, and analysis in R (49). Statistical analysis was performed using several tools: ape (v.5.8.1) was used in tree and sequence manipulation (50); Hmisc (v.5.2.3) (51) and vegan (v.2.6.10) (52) were used in advanced statistical analysis, data management, and correlation analysis. Bibliometrics were performed with rcrossref (v.1.2.0), with access to the CrossRef API for bibliographic metadata (53), and synthesisr (v.0.3.0) for search and deduplication of bibliographic references in systematic reviews (54). Data visualization was done with easyalluvial (v.0.3.2) for simplified creation of alluvial diagrams (55), ggplot2 (v.3.5.2) (56), ggpubr (v.0.6.0) (57), ggnewscale (v.0.5.1) (58), ggtext (v.0.1.2) (59), ggtree (v.3.14.0) (60), and ggtreeExtra (v.1.16.0) (61). LinkET (v.0.0.7.4) was used for visualization of correlations and associations between multiple variables (mantel test) (62), patchwork (v.1.3.0) in flexible combination and arrangement of multiple ggplot2 plots (63), pheatmap (v.1.0.12) in the generation of customizable heatmaps with clustering and annotation options (64), RColorBrewer (v.1.1.3) in predefined color palettes utilization (65), scales (v.1.4.0) in scale control, and transformations and labels in plots (66). Additionally, the Natural Map of Planet Earth was created using SCImago Graphica (v.31) (67). Improvement of vectorial figures was made using Affinity (v.2.6.3) (https://affinity.serif.com).

## Trait-based variable selection for MIRC guild identification

To identify MIRC guilds, relevant variables were selected through a combination of dimensionality reduction and correlation analysis. For dimensionality reduction, we applied (i) Kaiser's criterion to identify and retain key dimensions (eigenvalue >1); (ii) a contribution criterion requiring that the first three dimensions explain 70%–80% of the total variance; and (iii) a threshold criterion, where selected variables explain a percentage of variance greater than the average of the variables within the dimension ($1/p$, where $P$ is the number of variables). To minimize variable redundancy, a correlation analysis was performed. Variables showing strong collinearity were removed based on Spearman's $\rho > |0.7|$ with $P$ value of <0.05. Correlation strengths were categorized as weak $|0.1–0.3|$, moderate $|0.3–0.5|$, and strong $|0.5–1.0|$ using both monotonic (Spearman's $\rho$) and distance-based (Mantel's $\rho$) approaches.

## RESULTS

### First-order guild structure of iron redox microbiota and research trends

MIRC species (387 species) described in the global scientific literature (314 publications) were categorized into three first-order functional guilds: RED, OXI, and O/R. The RED guild was the most represented in both the number of distinct species (richness) and the number of published species descriptions, followed by the OXI and O/R guilds (Table 2). Most species were supported by molecular data (371 16S rRNA sequences and 293 genomes), underscoring the phylogenetic and functional anchoring of the guilds (Table S1). Globally, the MIRC guild members have been reported from 41 countries (Fig. 1A), with the highest contributions from the USA (21.5%), Japan (14.0%), Germany (8.3%), Russia (8.3%), and China (7.5%), while contributions of the global south remain limited.

Initial descriptions of species representatives of first-order MIRC guilds date back to 1949, with research in the field spanning 76 years (Table 2). The cumulative frequency of species descriptions reveals exponential growth for RED and OXI guilds, whereas the O/R guild displays a more linear trajectory (Fig. 1B). These patterns may stem from differences in sampling, detection, or culturing ease, prevailing research priorities, or the ecological rarity of habitats favoring O/R taxa. This meta-analysis encompassed contributions by 1,120 authors whose participation trends mirror those observed for guilds. Notably, a 3:1 ratio between author entries and newly described species suggests sustained and expanding research interest and activity in the field.

### Habitat and substrate associations shape the distribution of MIRC guilds

Habitat analysis (Fig. 1C) revealed that five types of environments (hot springs, marine systems, hydrothermal vents, acid mine drainage, and lakes) account for 46.8% of MIRC guild occurrences, with marine being the most frequent habitat for RED, acid mine drainages for OXI, and hot springs for O/R (Table S2a). Habitat occupancy was higher for RED and OXI than for O/R taxa, which were present in only 40% of habitats used by MIRC guild microbes (Table S2a). RED taxa were predominantly associated with particulate substrates across habitats (73% species), with high species richness in sediments from marine ($n = 18$), hot springs ($n = 11$), hydrothermal vents ($n = 17$), and lakes ($n = 12$). Also, a consistent presence of RED species across diverse anthropogenic settings (including compost, contaminated sites, industrial zones, mines, paddy fields, and wastewater) was observed, albeit at lower frequencies. Tight association of RED with particulate matter underscores their role in iron, metal, and organic matter turnover in oxygen-limited, metal- and/or carbon-rich microenvironments shaped by nature or human activity. Similar results were observed for OXI species, which inhabited preferentially particulate and fluid materials, yet also showed a strong presence in biofilms (13 species), reflecting metabolic flexibility and/or redox stratification within these systems or at their interfaces. O/R taxa, in turn, showed a more restricted use of source material (63%, Table S2b), most frequent in fluids and mineral sources. Hot springs, acid mine drainages, and other mine sites stood out as the habitat types supporting the largest numbers of members of all three first-order MIRC guilds (Table S2b), illustrating their role as ecological hotspots for Fe redox cycling. These systems likely offer the chemical diversity and redox gradients necessary for the co-existence of functionally distinct guilds.

**TABLE 2** Species count, temporal trends, and habitat associations of first-order MIRC guilds[a,b]

| First-order guild | Species count | Cumulative trend | First described representative | Predominant habitat association |
|---|---|---|---|---|
| RED | 239 | Exponential | *Desulfosporosinus orientis*, Singapore, 1965 | Particulate matter across hot springs, marine systems, hydrothermal vents, and lakes |
| OXI | 106 | Exponential | *Rhodomicrobium vannielii*, USA, 1949 | Fluids and biofilms in hot springs and acid mine drainage |
| O/R | 42 | Linear | *Sulfolobus acidocaldarius*, USA, 1972 | Biofilms, especially at redox interfaces |

[a]First-order guild alludes to broad, functionally defined groups of microbes that share a primary metabolic trait associated with a key ecological process in the MIRC or Fe wheel.
[b]O/R, dual-capacity Fe oxidizers/reducers; OXI, Fe(II) oxidizers; RED, Fe(III) reducers.

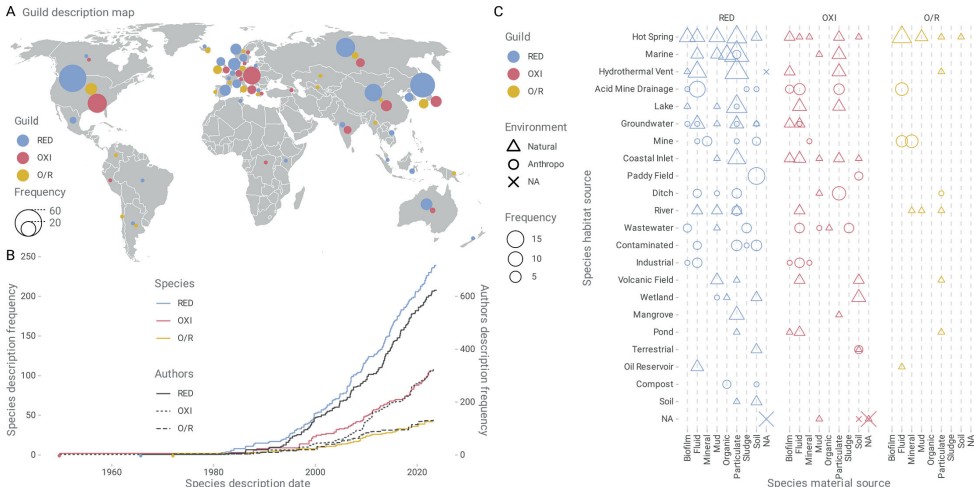

**FIG 1** Global distribution and bibliographic overview of MIRC guilds. (A) Guild description map. Natural earth projection was used. (B) Cumulative frequency in step visualization of new species and new authors over time. (C) Substrate association patterns of MIRC guilds across sampled environments, habitats, and material source shown as balloon plots illustrating the frequency distribution of RED, OXI, and O/R occurrences. MIRCs are colored as cyan blue, Fe(III)-reducing (RED); pink red, Fe(II)-oxidizing (OXI); and yellow, Fe-oxide/reducers (O/R).

## Functional trait syndromes shaping the ecology of first-order MIRC guilds

Evaluation of experimentally tested physiological traits of 371 MIRC species revealed distinct lifestyle patterns among and within first-order guilds (Fig. 2A; Table S3a). The RED guild exhibited the most uniform lifestyle profile, primarily characterized by species with chemotrophic metabolism ($n = 228$), organotrophic electron donors ($n = 160$), and heterotrophic carbon assimilation ($n = 188$). These organisms were largely anaerobic ($n = 152$) and neutrophilic ($n = 156$), with variations in their temperature regime preferences (73 psychrotolerant species and 55 mesophilic species). Salinity tolerances of this group were broad, though a relevant proportion remains untested ($n = 77$). The OXI guild (106 species) displayed broader metabolic diversity, a more heterogeneous metabolic profile of electron sources, including chemotrophy ($n = 99$), lithotrophy ($n = 42$), and mixed lithotrophy/organotrophy ($n = 58$), along with diverse carbon assimilation strategies (autotrophy: 43 species, heterotrophy: 41 species, and mixotrophy: 17 species). Taxa in this guild exhibited diverse oxygen requirements, including aerobic ($n = 32$), facultative ($n = 28$), and anaerobic ($n = 27$) lifestyles. Their environmental distribution spanned neutrophilic ($n = 71$) and acidophilic ($n = 15$) habitats, with temperature preferences ranging from psychrotolerant ($n = 38$) to mesophilic ($n = 27$) and exhibiting substantial variability in salinity tolerance. The O/R guild, though less represented (42 species), exhibited high metabolic versatility, combining lithotrophy and organotrophy ($n = 28$), linked to carbon mixotrophy ($n = 16$), heterotrophy ($n = 15$), and autotrophy ($n = 11$). These species predominantly adopted facultative ($n = 31$) or aerobic ($n = 8$) lifestyles and are strongly associated with extreme (EACI, 32 species) and hyperacidic (HACI, 4 species) environments. Thermal preferences of this guild spanned moderate thermophilic ($n = 10$), mesophilic ($n = 8$), and hyperthermophilic ($n = 5$) conditions. The most recurrent lifestyle configurations across guilds, interpreted here as trait syndromes (i.e., co-occurring sets of ecophysiological traits that reflect a guild's adaptation to specific environmental conditions) are shown in Table S3b. These were (i) for RED, taxa chemotrophic, organotrophic, heterotrophic, anaerobic, neutrophilic, mesophilic, and undefined salinity tolerance (11 species); (ii) for OXI, taxa chemotrophic, lithotrophic/organotrophic, heterotrophic, aerobic, neutrophilic, psychrotolerant, unspecified salinity (6 species); and (iii) for O/R, chemotrophic, lithotrophic/organotrophic, mixotrophic, facultative anaerobes, extreme acidophilic, hyperthermophilic, and unspecified

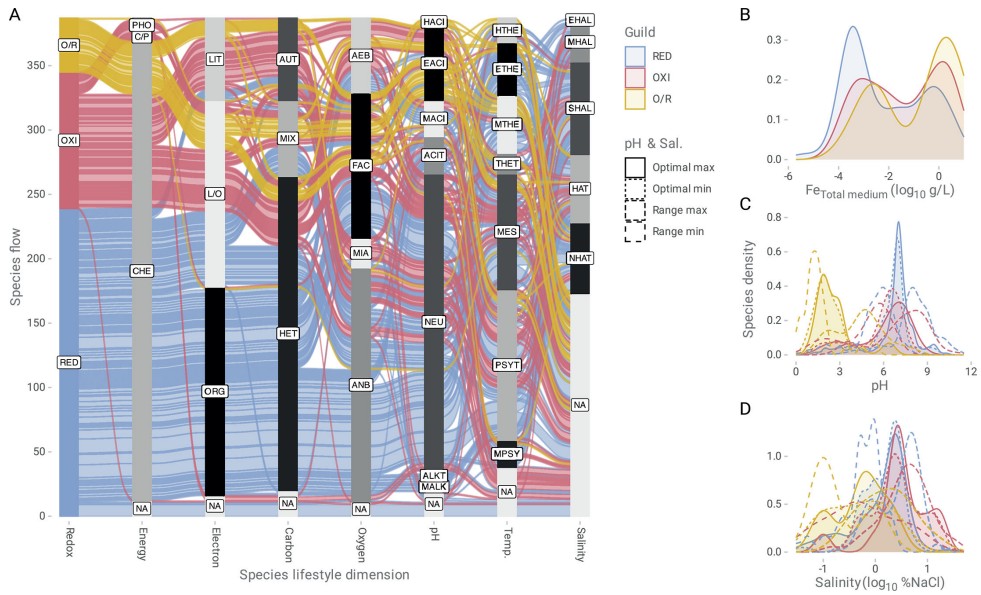

**FIG 2** Trait-based ecological profiling of MIRC guilds. (A) Distribution of alluvial flows of species according to lifestyle dimensions: redox activity in Fe(III)-reducing (RED, cyan blue), Fe(II)-oxidizing (OXI, pink red), and Fe-oxide/reducers (O/R, yellow); energy source (Energy) (chemotrophic [CHE], phototrophic [PHO], and chemo-phototrophic [C/P]); electron source (Electron) (organotrophic [ORG], litho-organotrophic [L/O], and lithotrophic [LIT]); carbon source (Carbon) (heterotrophic [HET], autotrophic [AUT], and mixotrophic [MIX]); oxygen tension (Oxygen) (anaerobic [ANB], facultative [FAC], aerobic [AEB], and microaerophilic [MIA]); pH distribution (pH) (hyperacidophilic [HACI]), extreme acidophilic [EACI], moderate acidophilic [MACI], acid-tolerant [ACIT], neutrophilic [NEU], alkali-tolerant [ALKT], and moderate alkaliphilic [MALK]); Temp. (hyperthermophilic [HTHE], extreme thermophilic [ETHE], moderate thermophilic [MTHE], thermotolerant [THET], mesophilic [MES], psychrotolerant [PSYT], and moderate psychrophilic [MPSY]); and salinity tolerances (Salinity) (extreme halophilic [EHAL], moderate halophilic [MHAL], slight halophilic [SHAL], halo-tolerant [HAT], and non-halo-tolerant [NHAT]). Unspecified species (NA). Kernel density distribution of physiological continuous traits (continuous histogram): (B) Total Fe concentration in the culture medium (log10 g/L), (C) pH, (D) Salinity (log10 % NaCl). Kernel density estimation was calculated with a Gaussian curve. The redox guilds were colored as cyan blue, Fe(III)-reducing (RED); pink red, Fe(II)-oxidizing (OXI); and yellow, Fe-oxide/reducers (O/R).

salinity (three species). Collectively, these recurring trait combinations reveal functional convergence within guilds (second-order guilds) and underscore their niche-specific contributions to iron redox cycling.

## Trait distributions underpin niche partitioning in MIRC guilds

Distinctive ecophysiological patterns among RED, OXI, and O/R guilds emerged from kernel density analyses of continuous traits recovered for MIRC taxa (Fig. 2B through D; Fig. S1). Bimodal distributions of total Fe concentration supplied in culture media during physiological characterization among RED, OXI, and O/R taxa were observed across guilds, reflecting distinct ecological strategies and physiological tolerances to Fe availability (Fig. 2B; Table 3). RED taxa media exhibited peaks at ~320 µg/L and ~320 mg/L, consistent with adaptation of iron reducers to both Fe-limited, particulate-rich environments and more Fe-replete reducing settings. For example, *Geobacter sulfurreducens* thrives under nanomolar Fe(III) concentrations by respiring poorly crystalline iron oxides using conductive pili and outer membrane cytochromes (68–70). A secondary group of RED taxa (e.g., *Desulfovibrio vulgaris*; 71, 72) is typical of anthropogenically influenced niches such as bioreactors and Fe-amended sediments with higher Fe availability. OXI taxa culture media showed a primary peak near ~2 g/L and a secondary at ~1 mg/L, reflecting the reliance of this guild representative on high concentrations of Fe(II) as energy source, typical of AMD and hydrothermal fluids (e.g., obligate iron

**TABLE 3** Bimodal iron concentration peaks across MIRC guilds reflecting trait-based niche specialization and/or metabolic versatility[a]

| Guild | Primary Fe peak (g/L) | Secondary Fe peak (g/L) | Ecological interpretation | Example |
|---|---|---|---|---|
| RED | ~0.00032 (−3.5 $\log_{10}$ g/L) | ~0.32 (−0.5 $\log_{10}$ g/L) | Adaptation to Fe-limited, particulate-rich, reducing environments | Low: *Geobacter sulfurreducens* High: *Desulfovibrio vulgaris* |
| OXI | ~2.0 (0.25 $\log_{10}$ g/L) | ~0.001 (−3.0 $\log_{10}$ g/L) | Adaptation to Fe-rich oxic environments (AMD, hydrothermal fluids) and Fe-limited oxic niches | High: *Leptospirillum ferrooxidans* Low: *Sideroxydans lithotrophicus* |
| O/R | ~3.2 (5 $\log_{10}$ g/L) | ~0.0032 (−2.5 $\log_{10}$ g/L) | Broad redox adaptability | High: *Acidithiobacillus ferrooxidans* (O) Low: *A. ferrooxidans* (R) |

[a]O/R, dual-capacity Fe oxidizers/reducers; OXI, Fe(II) oxidizers; RED, Fe(III) reducers.

oxidizer *Leptospirillum ferrooxidans*; 73, 74), as well as other somewhat Fe-limited oxic or microoxic niches (e.g., *Sideroxydans lithotrophicus* neutrophilic Fe-oxidizing microaerophiles inhabiting shallow subsurface aquifers or stream biofilms; 75, 76). O/R taxa culture media presented bimodal Fe concentration peaks at ~3.2 g/L and ~3.2 mg/L, spanning the broadest Fe concentration range among MIRC guilds. Peak values aligned with reported optimal values for iron-oxidizing *Acidithiobacillus* during iron oxidation (>2 g/L $Fe^{2+}$, 77) and reduction (~300 mg/L $Fe^{3+}$, 78), respectively. These profiles highlight metabolic plasticity and niche breadth across redox-dynamic habitats.

In terms of pH preferences, the RED and OXI guilds showed broader and higher optima (major peaks at 7.0–7.1), whereas the O/R guild was characterized by distinctly extreme acidic optima (major peak at 1.9), underscoring its niche specialization in low-pH environments (Fig. 2C). Salinity tolerance patterns further distinguished the guilds (Fig. 2D), with RED and OXI peaking at 2.5%–3.2% NaCl, with secondary peaks at both hypo- and hypersaline levels. In contrast, O/R guild members showed a tendency toward lower salinities (with peaks near 0.6%–0.8% NaCl), although this pattern should be interpreted cautiously, given that ~83% of taxa in this group lack salinity tolerance data. Slight differences also emerged from temperature data distributions, with RED and OXI displaying mesophilic optima (major peaks near 30°C) and O/R guilds showing higher thermal preferences (peak at 37.7°C), extending into potentially thermophilic ranges (Fig. S1). Observed distributions in environmental optima and ranges for these variables reflect broad adaptability within and across guilds while also highlighting distinct ecological specializations and niche differentiation among the RED, OXI, and O/R guilds.

## Phylogenetic diversity and functional trait distribution across MIRC guilds

MIRC guild species-level representatives ($n = 387$) spanned both the archaeal and bacterial domains of life (Fig. 3), distributed in 20 phyla and 42 classes (Table 4). The most frequent archaeal class was *Thermoprotei* ($n = 23$), and the most frequent bacterial classes were *Clostridia* and *Gammaproteobacteria*, with 47 species each, followed closely by *Desulfuromonadia* ($n = 43$). These three bacterial classes harbored taxa across the RED, OXI, and O/R guilds, underscoring their functional versatility in iron redox processes. The most represented genus was *Shewanella* (15 species), exclusively RED, thriving in neutral pH (6.0–7.5) and mesophilic conditions (7°C–37°C). Among acidophiles, the acidithiobacilli (seven species) showed high metabolic versatility: five were O/R, one OXI, and one RED, all extremely acidophilic (pH 2.0–2.5) and mesophilic to moderately thermophilic (29°C–45°C). Several taxonomic classes grouped species exclusively associated with a

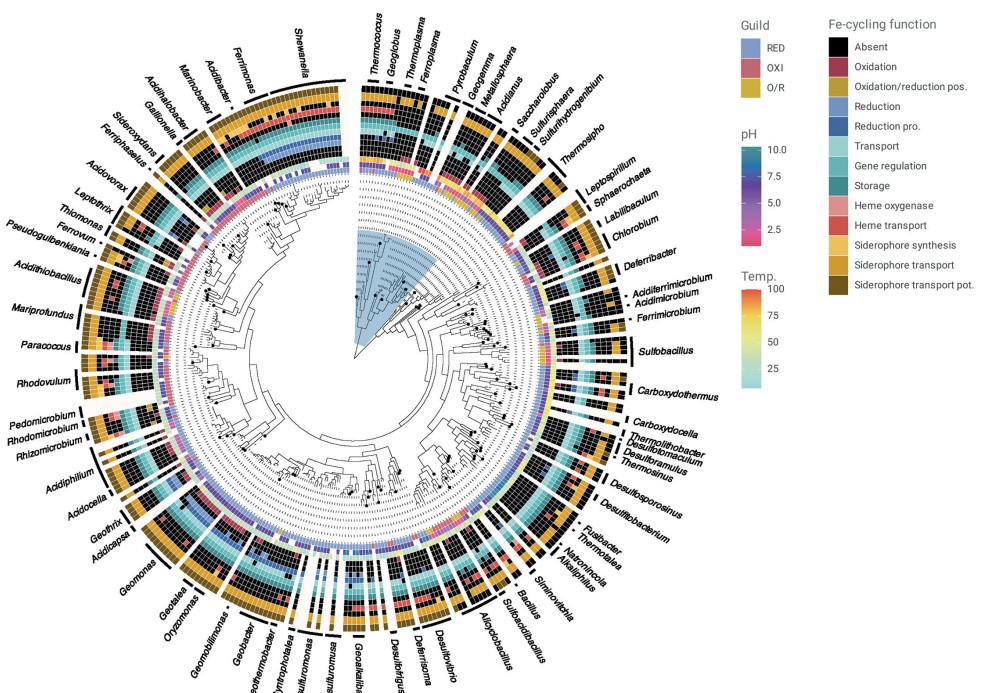

**FIG 3** Phylogenetic patterns and Fe-cycling functional traits through MIRC guilds. Maximum likelihood tree built with 371 16S rRNA sequences. 16S rRNA sequences were aligned with the L-INS-I method and analyzed by maximum likelihood (Tamura–Nei model). The tree was rooted with the Archaeal branch and is shown as a cladogram. In addition, genomic traits related to the iron cycle of 293 microorganisms were included. The blue mark on the tree indicates the species of the Archaea domain. The black circles at the tips of tree branches indicate unique taxa to their genus, known as holotype ($n$ = 115). Abbreviations: pos., possible; pot., potential; pro., probable (based on the traits defined for the Fe cycle in the FeGenie bioinformatics tool).

single MIRC guild: Fe(III)-reducing taxa ($n$ = 21 classes) or Fe(II)-oxidizing taxa ($n$ = 4 classes), typically with low species richness per class (one to eight species). Representative examples include *Deferribacteres* (six species) obligate or facultative anaerobes that reduce Fe(III) and other metals using $H_2$, acetate, or peptides in deep subsurface and hydrothermal environments, and *Nitrospira* (five species) obligate chemolithoautotrophs that oxidize Fe(II) under highly acidic and oxic conditions in AMD and hydrothermal settings.

At the functional level, genes associated with iron regulation, storage, Fe(II) uptake (e.g., *feo*B), and Fe(III)/siderophore transport were broadly distributed across the phylogeny (Fig. 3). In contrast, siderophore biosynthesis genes were rare and mostly restricted to a few neutrophilic heterotrophic or mixotrophic taxa within the OXI and RED guilds (only 10% of species). Heme transport genes were found in 34% of species, most commonly within the RED guild ($n$ = 68 out of 99 species with the trait). Conversely, siderophore transport systems were absent in 5% of species, the majority of which belonged to the O/R guild. Notably, four O/R acidophiles belonging to the genera *Acidiplasma*, *Ferroplasma*, and *Acidimicrobium* showed no known iron transport systems. Finally, 25% of the species lacked genome representatives (indicated as blank branches in Fig. 3), underscoring important gaps in the genomic landscape and highlighting areas of incomplete knowledge regarding iron-based energy metabolism and nutrient homeostasis in diverse iron-cycling microorganisms.

## Correlative patterns of phenotypic and genomic traits among MIRC guilds

Trait correlations (Spearman's $\rho$) and Mantel tests supported relevant patterns in the data analyzed (Fig. 4A). Across MIRC guilds, Fe oxidation capacity was positively associated with higher optimal Fe concentrations ($\rho$ = 0.35), with negative associations with pH

**TABLE 4** Taxonomic distribution of characterized MIRC guild members[a]

| Domain (n = 2) | Phylum (n = 20) | Class (n = 42) | No. of orders (n = 79) | No. of families (n =115) | No. of genera (n = 196) | No. of species (n = 387) | MIRC guild |
|---|---|---|---|---|---|---|---|
| Archaea | Candidatus Thermoplasmatota | *Thermoplasmata* | 1 | 3 | 4 | 8 | O/R, RED |
| | Thermoproteota | *Thermoprotei* | 3 | 3 | 9 | 23 | O/R, RED, OXI |
| | Thaumarchaeota | *Conexivisphaeria* | 1 | 1 | 1 | 1 | RED |
| | Methanobacteriota | *Methanomicrobia* | 1 | 1 | 1 | 1 | RED |
| | Euryarchaeota | *Archaeoglobi* | 1 | 1 | 2 | 5 | OXI, RED |
| | | *Methanomicrobia* | 2 | 2 | 2 | 2 | RED |
| | | *Thermococci* | 1 | 1 | 2 | 4 | RED, OXI |
| Bacteria | Acidobacteriota | *Holophagae* | 1 | 1 | 1 | 4 | RED |
| | | *Terriglobia* | 2 | 2 | 3 | 4 | RED |
| | | *Thermoanaerobaculia* | 1 | 1 | 1 | 1 | RED |
| | Actinomycetota | *Acidimicrobiia* | 1 | 1 | 7 | 7 | O/R, RED |
| | | *Actinomycetes* | 2 | 2 | 2 | 2 | RED |
| | | *Coriobacteriia* | 1 | 1 | 1 | 1 | RED |
| | Aquificota | *Aquificae* | 1 | 1 | 1 | 2 | O/R, RED |
| | Bacillota | *Bacilli* | 1 | 4 | 12 | 23 | OXI, RED, O/R |
| | | *Clostridia* | 8 | 16 | 31 | 47 | RED, O/R, OXI |
| | | *Negativicutes* | 2 | 2 | 4 | 4 | RED |
| | | *Thermolithobacteria* | 1 | 1 | 1 | 2 | RED |
| | | *Tissierellia* | 1 | 2 | 2 | 2 | RED |
| | Bacteroidota | *Bacteroidia* | 1 | 2 | 2 | 3 | RED, O/R |
| | | *Cytophagia* | 1 | 1 | 1 | 1 | RED |
| | Chlorobiota | *Chlorobiia* | 1 | 1 | 1 | 4 | OXI |
| | Chloroflexota | *Ardenticatenia* | 1 | 1 | 1 | 1 | RED |
| | Deferribacterota | *Deferribacteres* | 1 | 3 | 4 | 6 | RED |
| | Deinococcota | *Deinococci* | 1 | 1 | 1 | 1 | RED |
| | Ignavibacteriota | *Ignavibacteria* | 1 | 1 | 1 | 1 | RED |
| | Nitrospirae | *Nitrospira* | 1 | 1 | 1 | 5 | OXI |
| | Pseudomonadota | *Acidithiobacillia* | 1 | 1 | 2 | 7 | O/R, RED, OXI |
| | | *Alphaproteobacteria* | 4 | 9 | 16 | 38 | OXI, RED, O/R |
| | | *Betaproteobacteria* | 7 | 11 | 21 | 36 | OXI, RED |
| | | *Deltaproteobacteria* | 5 | 5 | 8 | 18 | RED, OXI |
| | | *Epsilonproteobacteria* | 1 | 1 | 1 | 1 | RED |
| | | *Gammaproteobacteria* | 8 | 14 | 18 | 47 | RED, OXI, O/R |
| | | *Zetaproteobacteria* | 2 | 2 | 2 | 7 | OXI |
| | Spirochaetota | *Spirochaetia* | 1 | 1 | 1 | 2 | RED |
| | Thermodesulfobacteriota | *Desulfobacteria* | 1 | 1 | 3 | 4 | RED, OXI |
| | | *Desulfobulbia* | 1 | 1 | 2 | 3 | RED |
| | | *Desulfovibrionia* | 1 | 1 | 4 | 6 | RED, OXI |
| | | *Desulfuromonadia* | 3 | 6 | 14 | 43 | RED, OXI, O/R |
| | | *Syntrophobacteria* | 1 | 1 | 1 | 1 | OXI |
| | | *Thermodesulfobacteria* | 1 | 1 | 1 | 1 | RED |
| | Thermotogota | *Thermotogae* | 2 | 3 | 3 | 8 | RED |

[a]MIRC, microbial iron redox cycle.

($\rho$ = −0.42/−0.46), Fe(II) ($\rho$ = −0.46) and complexed-Fe(III) transport (siderophore $\rho$ = −0.18, heme $\rho$ = −0.26), and Fe-dependent gene regulation ($\rho$ = −0.17). These results indicated that specialized acquisition or regulation systems (e.g., siderophores and heme transporters) are less prominent for OXI and O/R MIRC taxa (thriving at higher Fe concentrations), or alternatively less necessary under high Fe availability (as in acidic environments) than under limitation. Siderophore transport exhibited positive correlations with siderophore biosynthesis ($\rho$ = 0.33), consistent with the presence of cognate transporters in siderophore producers, regardless of guild. Also, with Fe storage ($\rho$ =

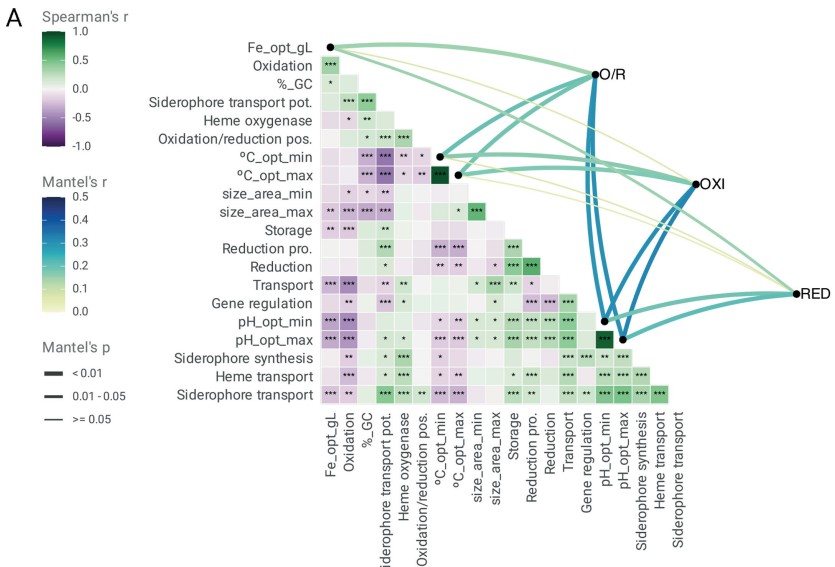

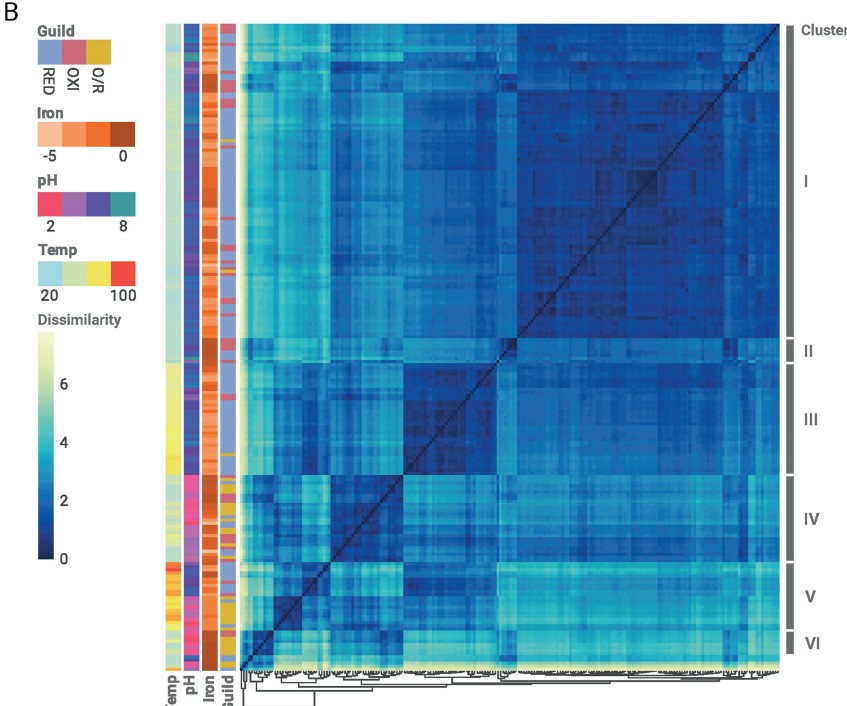

**FIG 4** Correlative and clustering patterns between phenotypic and genomic traits across microbial iron redox cycle guilds. (A) Correlogram showing monotonic (Spearman's $\rho$) and distance-based (Mantel's $\rho$) correlations between environmental optima (pH, temperature, and Fe concentration) and Fe-cycling genomic traits (12 Fe-related genomic traits inferred with FeGenie). Significant associations ($P < 0.05$) revealed moderate links between pH and genomic trait dissimilarity in OXI and O/R guilds, but weaker correlations for RED. (B) Hierarchical clustering of 208 representative taxa (subsampled from 387 species) based on 15 phenotypic and genomic variables resolved six clusters representing distinct ecological strategies shaped by environmental filtering and functional capacities. Additional details are provided as supplementary material in an extended figure legend (File_S1.doc). Abbreviations: %_GC, GC content; °C_opt_max, temperature optimal maximum; °C_opt_min, temperature optimal minimum; Fe_opt_gL, total Fe concentration in culture media; pH_opt_max, pH optimal maximum; pH_opt_min, pH optimal minimum; pos., possible; pot., potential; pro., probable (based on the traits defined for the Fe cycle in the FeGenie bioinformatics tool); size_area_max, microbial size area maximum; size_area_min, microbial size area minimum.

0.26), pointing to stronger pressures to retain iron intracellularly under conditions of low environmental availability or limitation, as experienced by RED taxa ($\rho$ = −0.10/−0.11) or diverse MIRC guild taxa under higher environmental pH ($\rho$ = 0.29–0.33). In turn, siderophore transport exhibited strong negative correlations with temperature optima ($\rho$ = −0.55), suggesting that Fe scavenging is disfavored at higher temperatures, likely reflecting the combined effects of environmental context (increased iron availability in fluids) and physicochemical constraints (potential limitations on siderophore stability or chelation efficiency at higher temperatures). Fe reduction was also negatively associated with temperature optima ($\rho$ = −0.16/−0.31), reflecting the ecological specialization of iron-reducing microbes for low-to-moderate temperature environments. According to these results, Fe scavenging via siderophores is favored in mesophilic oligotrophic contexts preferred by RED taxa. Optimal pH values exhibited a central role: positively associated with siderophore and Fe transport traits, and negatively with Fe oxidation and Fe concentration optima.

## Trait-based clustering suggests emergent exploitation patterns of MIRC guilds

Trait patterns were further supported by distance-based clustering, which resolved six main ecological groups among a representative subset of 208 MIRC taxa (Fig. 4B; Table S3b and c). Each cluster corresponds to a distinct combination of environmental preferences (pH and temperature) and iron cycling or homeostasis traits. Cluster I, the largest group, comprises mesophilic generalists with broad functional repertoires spanning both OXI and RED. Cluster II includes mesophilic taxa specialized in siderophore-mediated iron scavenging. Cluster III is dominated by thermophilic, neutrophilic RED enriched in iron transport and storage functions. Cluster IV encompasses acidophilic OXI and O/R taxa adapted to low-pH, Fe-rich environments. Cluster V groups thermophilic taxa that exhibit strong pH-driven niche partitioning between oxidative and reductive iron metabolisms. Finally, Cluster VI represents a small group of extreme acidophiles characterized by consistent iron oxidation capacity. These strong trait correlations and emergent clustering patterns reveal that Fe oxidation, reduction, and acquisition traits are not randomly distributed among MIRC taxa genomes but are rather tightly linked to organismal ecological preferences and adaptive potential, constituting second-order MIRC guilds.

## DISCUSSION

### Novelty and significance of the MIRC guild framework

This study offers a global-scale, trait-based synthesis of microbial iron redox cycling, integrating nearly 400 species studied over more than seven decades into a structured guild framework that advances our understanding of a biogeochemically critical yet often cryptic process. By delineating first-order guilds based on core functional roles, Fe(II) oxidation, Fe(III) reduction, and reversible Fe cycling, rather than phylogenetic affiliation, the framework decouples function from taxonomy, aligning microbial traits with ecosystem processes. This is particularly relevant for iron cycling, where multiple oxidation and reduction pathways frequently co-occur in tandem within the same environment (3, 79) and where functional redundancy among phylogenetically distant taxa can obscure ecological patterns (80).

Crucially, the incorporation of second-order guilds captures fine-grained distinctions in trait syndromes, such as acidophily, metabolic breadth, or redox flexibility, enabling us to formalize ecophysiological niches that would otherwise be collapsed in a simple oxidizer/reducer dichotomy. For instance, acidophilic Fe(II) oxidizers occupy chemically distinct redox niches compared to neutrophilic nitrate-dependent or photoferrotrophic oxidizers, despite catalyzing the same nominal transformation. Similarly, Fe reducers range from anaerobic respiratory generalists like *Geobacter* spp. to acidophilic taxa employing alternative, less understood, biochemical pathways (21).

By embracing this layered guild structure, a conceptual scaffold that mirrors other elemental cycles—e.g., the partitioning of nitrifiers and denitrifiers in nitrogen cycling—allowing better integration of MIRC into ecosystem and biogeochemical models (31) and cross-system generalizations is provided. This approach also responds to recent calls for functional trait-based classification systems in microbial ecology that can accommodate both metabolic potential and ecological behavior (81, 82). This framework demonstrates how such a perspective can unlock new insights into functional convergence, divergence, and environmental filtering in MIRC communities.

As with any meta-analysis, the patterns and correlations identified here are necessarily constrained by the current state of knowledge. This limitation is particularly relevant for microbial iron redox metabolisms, which remain less comprehensively characterized at the physiological and mechanistic levels than other major respiratory guilds (e.g., sulfate reducers, nitrate reducers, or methanogens). Consequently, the guild structure and trait correlations presented in this study should be viewed as a synthesis of available evidence rather than a definitive or exhaustive representation. Ongoing discovery of novel taxa, refinement of metabolic annotations, and improved physiological characterization (especially for underexplored environments and non-model organisms) are likely to expand, refine, or reshape these relationships.

## Ecophysiological trait syndromes and niche partitioning in iron cycling

The guild-level analysis performed uncovered consistent trait syndromes, i.e., clusters of physiological traits that tend to co-occur (83–86), shaping the ecological niches of iron-cycling microbes. Divergent adaptations to iron availability, pH, and thermal regime emerged as central axes separating oxidizing and reducing taxa, complementing known environmental drivers like oxygen and redox gradients (3, 87).

As illustrated in Fig. 5, oxidizers, reducers, and dual-capacity taxa distribute predictably along combined gradients of pH and oxygen availability, reflecting how redox potential and acidity jointly structure the ecological space of MIRC communities and how trait syndromes align with these environmental filters. Fe(III) reducers examined in this study predominantly originated from anaerobic environments (ANB > FAC >> AEB), where soluble iron is scarce and insoluble ferric oxides serve as the terminal electron acceptors. This group included classical dissimilatory Fe(III)-reducing bacteria from neutral pH sediments (e.g. *Shewanella* and *Geobacter*), which are characterized by specialized traits including multiheme c-type cytochromes, extracellular electron shuttles, and conductive appendages (e.g., nanowires) that facilitate direct electron transfer to solid-phase Fe(III) minerals (88–91). It also encompasses more specialized microorganisms, such as *Natronincola ferrireducens* from soda lake sediments, which can reduce amorphous ferric hydroxide to magnetite under saline conditions, although the underlying mechanisms remain unresolved (92). Many sediment-dwelling Fe-reducers exhibit broad flexibility in potential electron donors (ORG > O/L> LIT) and alternative electron acceptor usage, often switching to metals, anodes, sulfate, or humic substances when Fe(III) is depleted, reflecting a generalist respiratory strategy (93). As emphasized by Lovley et al. (16), such respiratory versatility is not incidental but represents a key ecological adaptation, enabling these organisms to persist and outcompete others in fluctuating anoxic environments, particularly within sediments and subsurface systems.

Among acidophilic RED taxa (e.g., *Acidiphilium* and *Ferroplasma*), genomic evidence indicates the use of rewired or repurposed iron-oxidation pathways to support Fe(III) reduction under acidic conditions, effectively bypassing the canonical systems that characterize known neutrophilic dissimilatory Fe(III) reducers (21). These adaptations likely reflect the distinct redox chemistry and energetic constraints of low-pH environments, where standard extracellular electron transfer mechanisms seem impaired (94, 95). It is worth noting that all tested acidophilic prokaryotes capable of oxidizing Fe(II) also retain the ability to reduce Fe(III), reflecting the reversibility of their iron redox systems under shifting oxic–anoxic conditions (23, 96). The reverse, however, is not true, as many acidophilic heterotrophs (e.g., *Acidiphilium* spp.) exclusively reduce Fe(III)

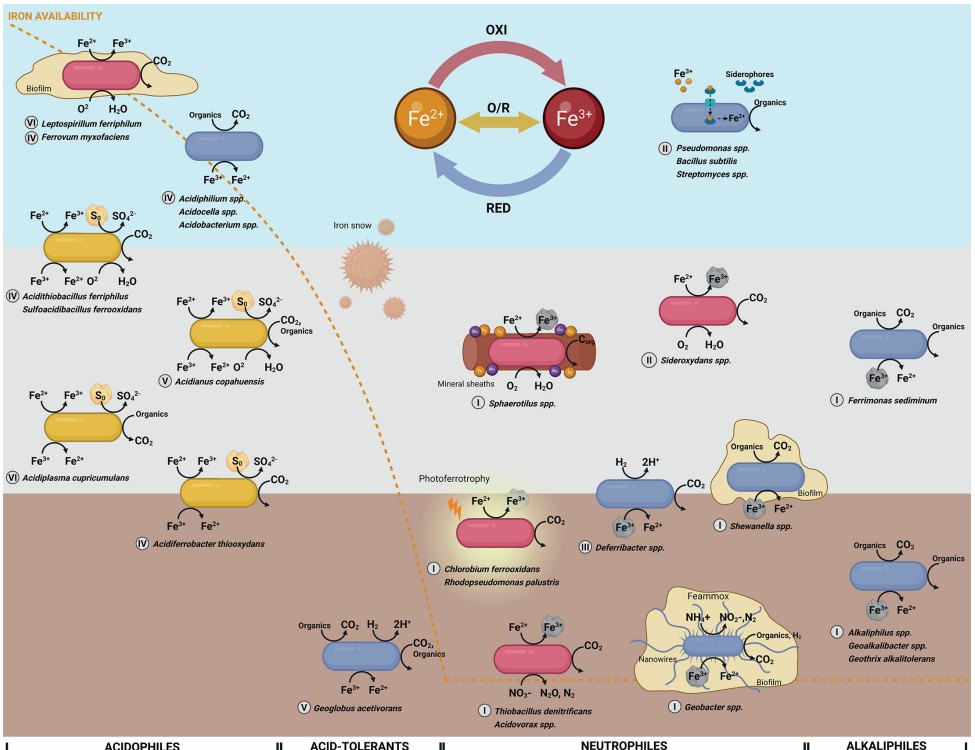

**FIG 5** Conceptual model of MIRC guilds along oxygen and pH gradients. The diagram illustrates the distribution of first-order guilds (oxidizers in red, reducers in blue, and oxidizers/reducers in yellow) across environmental gradients of pH (*X*-axis, separated by tolerance) and oxygen (*Y*-axis, aerobic in light blue, microoxic in gray, and anaerobic in brown). Arrows represent major biogeochemical reactions mediated by each guild. $Fe^{3+}$ is depicted as either soluble or associated with solid Fe(III)-oxihydroxides (rock symbol). Elemental sulfur ($S^0$) is represented by a yellow rock. The orange dashed line indicates the overall availability of iron across the pH gradient. Greek numerals next to taxon names indicate their MIRC cluster assignation. Representative taxa are shown within each niche, including acidophilic iron oxidizers (*Leptospirillum* spp. and *Ferrovum myxofaciens*), acidophilic reducers (*Acidobacterium* spp., *Acidocella* spp., and *Acidiphilum* spp.), acidophilic iron oxidizer/reducers coupled with sulfur oxidation (*Acidianus copahuensis*, *Acidiferrobacter thiooxydans*, *Acidiplasma cupricumulans*, *Acidithiobacillus ferriphilus*, and *Sulfoacidibacillus ferrooxidans*), acid-tolerant reducers (*Geoglobus acetivorans*), neutrophile nitrate-dependent oxidizers (*Thiobacillus denitrificans* and *Acidovorax* spp.), photoferrotrophs (*Chlorobium ferrooxidans* and *Rhodopseudomonas palustris*), siderophore-producing scavengers (*Pseudomonas* spp., *Bacillus subtilis*, and *Streptomyces* spp.), biofilm-associated reducers (*Shewanella* spp. and *Geobacter* spp. [capable of Feammox and nanowire-mediated electron transfer]) and oxidizers (*Sphaerotilus* spp. [mineral sheath formation] and *Sideroxydans* spp.), thermophilic or mixotrophic reducers (*Deferribacter* spp. and *Geoglobus acetivorans*), alkali-tolerant reducers (*Ferrimonas sediminum*), and alkaliphilic reducers (*Alkaliphilus* spp., *Geoalkalibacter* spp., and *Geothrix alkalotolerans*).

without the capacity for Fe(II) oxidation. Furthermore, the predominance of soluble Fe(III) species at pH below 3 reduces the need for contact-dependent reduction strategies (13, 21). Notably, several acidophilic species are known to engage in indirect iron reduction via sulfur disproportionation, producing $H_2S$ that abiotically reduces Fe(III), a strategy absent in neutrophiles (97, 98). These unconventional mechanisms provide a plausible explanation for why numerous genomes within this guild lack the expected genetic hallmarks of iron respiration (most of which are either pH or temperature extremophiles). Instead, extremophilic Fe reducers appear to have evolved alternative routes tailored to habitats of extreme pH or temperature, underscoring the ecological and mechanistic diversity of iron reduction.

Fe(II) oxidizers exhibited greater ecological versatility, spanning habitats of broader iron concentrations and multiple "subniches" via aerobic, anaerobic, phototrophic, and nitrate-reducing strategies consistent with previous knowledge (3, 80, 99,

100). Neutrophilic photoferrotrophs and nitrate-dependent oxidizers possess narrower metabolic repertoires adapted to redox-stratified habitats or redox transition zones, often exhibiting guild-specific traits like bacteriochlorophyll (101) or denitrification enzymes (102). Importantly, despite their inclusion within the Fe(II)-oxidizing guild, the energetic basis of nitrate-dependent Fe(II) oxidation remains unresolved (103). While nitrate reduction is unequivocally linked to energy conservation in these organisms, it has not been definitively demonstrated that Fe(II) oxidation itself contributes directly to cellular energy generation. Different studies suggest that Fe(II) oxidation may proceed indirectly via reactive nitrogen intermediates such as nitrite, which can abiotically oxidize Fe(II) under anoxic conditions (3, 12, 104).

This contrasts sharply with acidophilic Fe(II) oxidizers, in which iron oxidation is unequivocally coupled to chemolithotrophic energy conservation. Acidophilic Fe(II) oxidizers (e.g., *Acidithiobacillus* and *Leptospirillum*) thrive at pH <3, where ferrous iron is stable and abundant; in these settings, iron is not a limiting nutrient but rather an energy source available in millimolar concentrations (96, 105). Acidophilic iron oxidizers display traits such as acid tolerance, metal resistance, and matrix-mediated evasion of Fe(III) precipitation (106, 107) that enable their survival under iron- and metal-rich conditions. These adaptations underscore the metabolic and environmental specialization encoded in OXI subguilds, with fundamentally distinct metabolic architectures and trait syndromes shaped by the energy landscape and chemical speciation of their niches. In this context, second-order guild classifications effectively capture meaningful ecological and mechanistic differentiation beyond the broad Fe-oxidizer label. Nevertheless, it is important to recognize that iron redox cycling in natural environments also includes indirect oxidation processes mediated by microbial metabolic by-products such as nitrite, sulfide, or reactive oxygen species, even under anaerobic or microaerophilic conditions. Although these transformations fall outside the scope of the trait-based guild framework used here, owing to their lack of direct linkage to conserved iron redox metabolisms, they likely contribute to overall iron turnover and may modulate biologically mediated pathways.

Importantly, the analysis also identified a set of dual-capability O/R guild members that defy a simple oxidizer/reducer categorization. Notably, prior biogeochemical models rarely accounted for organisms capable of both reactions. These taxa (e.g., *Acidithiobacillus* spp. and *Acidiferrimicrobium australe*) can both oxidize Fe(II) and reduce Fe(III) under different conditions (108, 109). For instance, several iron-oxidizing acidithiobacilli (e.g., *A. ferrooxidans*, *A. ferridurans*, and *A. ferrivorans*) are known for their ability to oxidize iron and sulfur under aerobic conditions, yet some strains can switch to anaerobic growth by using ferric iron as an electron acceptor (with elemental sulfur or hydrogen as donor), effectively behaving as an iron reducer (21). Another interesting example is *Metallosphaera sedula*, which exhibits this dual iron redox capability, aerobically oxidizing Fe(II) or sulfur and anaerobically (or microaerobically) reducing Fe(III) while thriving under extreme thermophilic and acidophilic conditions (110). The guild framework formally groups these versatile organisms, highlighting a trait syndrome of reversible iron redox cycling (dual-capacity Fe oxidizers/reducers). Several members of this group possess complete sets of both iron-oxidation and iron-reduction machinery in their genomes, coupled with regulatory networks that enable rapid switching between these modes in response to environmental cues. However, many O/R organisms characterized to date tend to lean toward either an OXI-like or RED-like genomic repertoire, with reversible iron cycling emerging from partial pathway overlap rather than specialized systems (pathway reversibility; e.g., see reference 21) or presently undefined mechanisms. Notably, most acidophilic Fe-metabolizing prokaryotes fall within this category, whereas obligate oxidizers such as *Leptospirillum* and *Ferrovum* represent exceptions. To date, no neutrophilic Fe(II) oxidizer has been shown to perform reciprocal Fe(III) reduction, underscoring a key physiological distinction among neutrophilic and acidophilic taxa, yet also the geochemical permissiveness of low-pH systems, where both Fe(II) and Fe(III) remain soluble and bioavailable. Ecologically, the metabolic flexibility of O/R guild

members likely provides a competitive advantage with respect to separate oxidizer and reducer communities that helps sustain cryptic iron cycling by continual regeneration of Fe(III)/Fe(II) (3).

In addition to Fe utilization as electron donor or acceptor, acquisition and homeostasis strategies also partitioned differentially among MIRC guilds. Our analysis reveals that siderophore production, an energetically costly but high-affinity strategy for Fe(III) scavenging, is unevenly distributed across guilds, reflecting habitat-specific iron availability. Siderophore biosynthesis is prevalent among heterotrophic taxa inhabiting iron-limited environments (e.g., marine or oxic soils), whereas anaerobic Fe reducers from iron-rich niches (e.g., *Geobacter* and *Desulfovibrio*) typically lack biosynthetic genes but retain uptake systems, suggesting reliance on exogenous siderophores. This supports a model of cooperative or exploitative iron acquisition, in which many MIRC members import xenosiderophores via TonB-dependent receptors without incurring the biosynthetic cost of production. Conversely, acidophilic obligate Fe(II) oxidizers and reversible Fe cyclers invariantly lacked siderophore biosynthetic genes, consistent with the greater availability of soluble iron in their habitats. Instead, these taxa prioritize mechanisms for iron storage (e.g., bacterioferritin), homeostasis regulation (e.g., Fur/DtxR), and redox exploitation through expanded Fe(II) oxidation gene repertoires. Tight control of intracellular iron, together with rapid extracellular Fe(II) oxidation, has actually been proposed as part of a broader tolerance strategy to cope with millimolar ferrous iron concentrations (20, 111).

Collectively, these results demonstrate that iron-cycling microbes deploy coordinated trait strategies and genomic investments aligned with environmental constraints, spanning a continuum from "iron hunters" in oligotrophic niches to "iron tolerators" in metal-rich systems. The Guild Exploitation Pattern is proposed as a conceptual model that captures this distribution of trait hypervolumes across guilds, emphasizing how metabolic flexibility, environmental filtering, and ecological trade-offs structure the MIRC community. The Guild Exploitation Pattern concept provides a unifying lens to interpret how guilds partition ecological space and dynamically respond to redox fluctuations, offering predictive insight into microbial contributions to iron cycling under changing environmental conditions.

## Temperature constraints on iron redox metabolisms

Temperature is a fundamental determinant of microbial distribution and physiology, and the MIRC microbiome is no exception; our analysis indicated that different iron guilds are not uniformly distributed across the thermal spectrum. Trait-based analysis revealed that most known iron-cycling microorganisms are mesophiles, with optimal growth between 20°C and 40°C, and near-neutral pH, consistent with trends noted in previous surveys of Fe-reducing strains (e.g., see reference 94), showing that while iron reduction is theoretically favorable over a wide pH–temperature space, it has only been observed in culture at either extreme pH or extreme temperature. The MIRC guilds database, which spans diverse environments, similarly shows that thermophilic or psychrophilic iron cyclers (10 RED, 1 O/R) remain rare or underreported. MIRC mesophiles were found from pH 2 to 8, and thermophiles in neutral to mildly acidic conditions in diverse habitat types, but very few neutrophilic hyperthermophilic Fe reducers ($n = 7$, Archaea: 3 species of *Pyrobaculum*, *Geoglobus ahangari*, and *Thermococcus siculi*; Bacteria: *Geothermobacterium ferrireducens* and *Thermotoga maritima*), fewer neutrophilic hyperthermophilic ($n = 1$, Bacteria: *Ferroglobus placidus*) or thermophilic Fe oxidizers ($n = 1$, Bacteria: *Sulfurihydrogenibium azorense*), and no obligate psychrophilic Fe oxidizer in culture validly described to date (only a few RED: *Shewanella arctica*, *Maridesulfovibrio frigidus*, and *Geopsychrobacter electrodiphilus*). Thermophilic iron oxidation was mostly confined to acidophilic archaeal lineages (ETHE: *Acidianus*, *Sulfolobus*, *Sulfuracidifex*, and *Metallosphaera*; HTHE: *Palaeococcus*, *Saccharolobus*, and *Sulfurisphaera*), which invariantly inhabit hot acidic springs. According to environmental metagenomic studies (112), cold-adapted iron metabolizers are likely active in permafrost, alpine soils, or deep ocean waters,

reflecting sampling and research biases. Underrepresentation of these groups may reflect genuine geochemical and physiological constraints on iron redox chemistry under hot or cold conditions, where iron speciation and stability limit the feasibility of microbial iron cycling. Alternatively, "empty niches" in the iron cycle could represent unexplored biomes in need of targeted exploration (e.g., hydrothermal vents and deep hypersaline brines) to expand the known limits of microbial iron cycling, both for understanding life's adaptability and informing biotechnological applications in extreme settings.

## Research biases, knowledge gaps, and future directions

The trait-based synthesis of MIRC guilds has provided relevant insights into the functional ecology of iron-metabolizing microbes. However, it also exposed several critical research biases and knowledge gaps that constrain the understanding of the diversity, distribution, and ecological roles of these organisms. Addressing these limitations is essential to advance both theoretical and applied perspectives on the microbial iron cycle.

Iron cycling research has historically relied on a small group of well-studied organisms that dominate both the literature and genomic databases, such as *Geobacter metallireducens*, *Shewanella oneidensis*, *L. ferrooxidans*, and *Acidithiobacillus ferrooxidans*. This "model organism bias" has shaped paradigms around particular physiological strategies, such as conductive nanowires for Fe(III) reduction in *Geobacter* (e.g., see reference 69) or oxidation pathways in *Acidithiobacillia* acidophiles (e.g., see reference 113), while potentially overlooking alternative mechanisms employed by less-characterized taxa (e.g., soluble electron shuttles in *Acidobacteria* or fermentative Fe reducers in *Firmicutes*). Compounding this issue is the uneven availability of genomic and phenotypic data across MIRC taxa, and the geographic and habitat sampling biases. Some organisms have complete genomes and detailed physiological characterizations, while others are known only from 16S rRNA sequences or a few enrichment cultures. Mis-annotations are common, with iron-related genes frequently labeled as "hypothetical proteins" or "cytochromes" without functional validation. Research focus has left large geographic regions (e.g., Africa, Southeast Asia, and South America) and major ecosystem types (e.g., tropical wetlands, lateritic soils, permafrost, and the deep subsurface) underrepresented in the known diversity of MIRC taxa. The presented data set reflects this skew, with few isolates from iron-rich regions of the global South despite environmental DNA studies suggesting widespread iron cycling activity in these areas (114–116). Whether this pattern reflects true ecological differences or simply research effort disparities remains unresolved, but the bias highlights an urgent need for more inclusive biogeographic sampling.

Many knowledge gaps also stem from biases in experimental design. Traditional media formulations and enrichment protocols have favored organisms that grow on common substrates (e.g., acetate for Fe reducers and $FeSO_4$ for oxidizers), potentially excluding microbes with alternative metabolisms. Rare or condition-dependent processes, such as anaerobic ammonium oxidation coupled to Fe(III) reduction, remain poorly understood, largely because typical culturing approaches do not combine the necessary electron donors and acceptors. Similarly, recent discoveries of nitrate-dependent Fe oxidizers and marine Fe mat-formers (e.g., *Zetaproteobacteria*) have emerged only through targeted efforts in underexplored habitats, highlighting the importance of adapting methodologies to novel ecological contexts. In this light, iron concentration traits derived from culture media formulations must also be interpreted cautiously. Media recipes are frequently inherited across studies and often provide iron in excess of physiological demand, reflecting experimental convention rather than biological optima. These practices obscure finer-scale adaptations to iron availability and limit the resolution with which trait distributions can be interpreted.

Taxonomy-based classifications often fail to capture ecological function, leading to inconsistencies between what microbes are named and what they do. This guild-based approach revealed that certain traits, such as nitrate-dependent Fe oxidation

or reversible switching between oxidation and reduction, are phylogenetically widespread but inconsistently recognized in biogeochemical models. For example, organisms capable of both oxidizing and reducing iron (O/R guild) have been largely overlooked in conceptual models of iron cycling, despite their potential to stabilize redox fluxes in fluctuating environments. These inconsistencies are not just semantic but also limit the ability to predict microbial responses to environmental change.

In conclusion, the MIRC guild framework provides a novel, trait-based lens to decode the complexity of microbial iron cycling, moving beyond taxonomic constraints to capture functional convergence, divergence, and niche specialization. By structuring iron metabolisms into first- and second-order guilds, it bridges microbial traits with ecosystem processes, offering a predictive tool for understanding how environmental gradients shape iron redox dynamics. This approach not only highlights overlooked metabolic strategies and ecological roles but also sets the stage for targeted exploration of underrepresented lineages, habitats, and biogeochemical interactions. As such, the MIRC guild framework represents a conceptual advance with broad relevance for integrating microbial iron cycling into global models of elemental fluxes and ecosystem functioning.

## ACKNOWLEDGMENTS

This work builds upon 76 years of research and the contributions of more than a thousand scientists whose collective efforts laid the foundation for this synthesis. The authors gratefully acknowledge support from Agencia Nacional de Investigación y Desarrollo (ANID)/BASAL/FB210008, ANID/FONDECYT 1221035 (R.Q.), and ANID/FONDECYT 3230527 (F.I.) grants. Additional support was provided by doctoral scholarships ANID/21241467 (C.R.-V.) and ANID/21241350 (S.R.-I.) and by the Vicerrectoría de Investigación y Doctorados of Universidad San Sebastián through PhD scholarship 10202955 (F.D.-G.).

Funders: ANID (grant nos. BASAL/FB210008 and FONDECYT 1221035 [R.Q.], FONDECYT 3230527 [F.I.], ANID/PhD Scholarship 21241467 [C.R.-V.], and ANID/PhD Scholarship 21241350 [S.R.-I.]) and Vicerrectoría de Investigación y Doctorados Universidad San Sebastián (grant no. PhD scholarship 10202955 [F.D.-G.]).

## AUTHOR AFFILIATIONS

[1]Centro Científico y Tecnológico de Excelencia Ciencia & Vida, Santiago, Chile

[2]Programa de Doctorado en Biotecnología y Bioemprendimiento, Facultad de Medicina, Universidad San Sebastián, Santiago, Chile

[3]Programa de Doctorado en Biología Computacional, Facultad de Ingeniería, Arquitectura y Diseño, Universidad San Sebastián, Santiago, Chile

[4]Centro GEMA | Genómica, Ecología & Medio Ambiente, Universidad Mayor, Santiago, Chile

[5]Institute of Biosciences, Technische Universität Bergakademie Freiberg, Freiberg, Germany

[6]School of Environmental and Natural Sciences, Bangor University, Bangor, United Kingdom

[7]Faculty of Health and Life Sciences, Coventry University, Coventry, United Kingdom

[8]Natural History Museum, London, United Kingdom

[9]Centro Regional Universitario Bariloche-UNComahue, Instituto de Investigaciones en Biodiversidad y Medioambiente (INIBIOMA), CCT-Patagonia Norte, CONICET, San Carlos de Bariloche, Argentina

[10]Facultad de Ciencias, Universidad San Sebastián, Santiago, Chile

## AUTHOR ORCIDs

Fernando Díaz-González  http://orcid.org/0000-0001-5397-6179
Camila Rojas-Villalobos  http://orcid.org/0000-0003-2228-342X

Francisco Issotta ⬤ http://orcid.org/0000-0001-6414-1935
Sofía Reyes-Impellizzeri ⬤ http://orcid.org/0000-0002-0905-6508
Sabrina Hedrich ⬤ http://orcid.org/0000-0001-6125-5566
D. Barrie Johnson ⬤ http://orcid.org/0000-0001-7348-4016
Pedro Temporetti ⬤ http://orcid.org/0009-0005-3052-2023
Raquel Quatrini ⬤ http://orcid.org/0000-0003-2600-2605

## FUNDING

| Funder | Grant(s) | Author(s) |
|---|---|---|
| Agencia Nacional de Investigación y Desarrollo | ANID/BASAL FB210008 | Raquel Quatrini |
| Agencia Nacional de Investigación y Desarrollo | ANID/FONDECYT 1221035 | Raquel Quatrini |
| Agencia Nacional de Investigación y Desarrollo | ANID/FONDECYT 3230527 | Francisco Issotta |
| Agencia Nacional de Investigación y Desarrollo | ANID/BECAS 21241467 | Camila Rojas-Villalobos |
| Agencia Nacional de Investigación y Desarrollo | ANID/Becas 21241350 | Sofía Reyes-Impellizzeri |
| Vicerrectoria de Investigacion y Doctorados Universidad San Sebastian | Beca 10202955 | Fernando Díaz-González |

## AUTHOR CONTRIBUTIONS

Fernando Díaz-González, Formal analysis, Investigation, Methodology, Validation, Visualization, Writing – original draft | Camila Rojas-Villalobos, Investigation, Methodology | Francisco Issotta, Investigation, Methodology | Sofía Reyes-Impellizzeri, Validation, Visualization | Sabrina Hedrich, Investigation, Validation, Writing – review and editing | D. Barrie Johnson, Investigation, Validation, Writing – review and editing | Pedro Temporetti, Conceptualization, Formal analysis, Supervision, Writing – original draft, Writing – review and editing | Raquel Quatrini, Conceptualization, Investigation, Project administration, Resources, Supervision, Writing – original draft, Writing – review and editing

## DATA AVAILABILITY

The gene and genome sequences analyzed in this study can be found in the online NCBI repository, with the accession numbers listed in Table S1.

## ADDITIONAL FILES

The following material is available online.

Supplemental Material

**Figure S1 (mSystems01488-25-s0001.pdf).** Morphological and taxonomic description of MIRC guilds.
**Supplemental Material (mSystems01488-25-s0002.docx).** Extended legend for Fig. 4, supplemental figures, and supplemental tables captions.
**Table S1 (mSystems01488-25-s0003.xlsx).** Microbial iron redox cycle representatives included in the metanalysis.
**Table S2 (mSystems01488-25-s0004.xlsx).** Habitat distribution and source material of first-order microbial iron redox cycle guilds.
**Table S3 (mSystems01488-25-s0005.xlsx).** Trait syndromes representing the most recurrent lifestyle configurations within first-order MIRC guilds.

Open Peer Review

**PEER REVIEW HISTORY (review-history.pdf).** An accounting of the reviewer comments and feedback.

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
