## [Reviewer comments · mSystems]

Trait-Based Meta-Analysis of Microbial Guilds in the Iron Redox Cycle

Fernando Díaz-González, Camila Rojas Villalobos, Francisco Issotta, Sofia Reyes Impellizzeri, Sabrina Hedrich, D. Barrie Johnson, Pedro Temporetti, and Raquel Quatrini

Corresponding Author(s): Raquel Quatrini, Facultad de Medicina y Ciencia, Universidad San Sebastián, Santiago, Chile

Review Timeline:

Submission Date:	October 16, 2025
Editorial Decision:	December 11, 2025
Revision Received:	December 17, 2025
Accepted:	December 30, 2025

Editor: Marcela Hernandez

Reviewer(s): Disclosure of reviewer identity is with reference to reviewer comments included in decision letter(s). The following individuals involved in review of your submission have agreed to reveal their identity: Ricardo Amils (Reviewer #1); Michael Christopher Macey (Reviewer #4)

Transaction Report:

DOI: <https://doi.org/10.1128/msystems.01488-25>

Re: mSystems01488-25 (**Trait-Based Meta-Analysis of Microbial Guilds in the Iron Redox Cycle**)

Dear Dr. Raquel Quatrini:

As you can see from the reviewers' comments, they all agree that your review will provide valuable knowledge to the scientific community working on the iron cycle. Please carefully address each of the reviewers' comments before submitting a revised version.

Revision Guidelines

Sincerely,
Marcela Hernandez
Editor
mSystems

Reviewer #1 (Comments for the Author):

The ms "Trait-Based Mera-Analysis of Microbial Guilds in the Iron Redox Cycle" by Díaz González et al., is an interesting work concerning a fundamental redox reaction in nature, the Fe cycle. The work deals with an outstanding systematic work using most of the information concerning the subject and the conclusions are very interesting for microbial ecologists. Some suggestions

and comments:

- Although is not the aim of the work nor the methodologies used, it might be interesting to mention that there is an important level of oxidation of Fe, mainly in anaerobic conditions, produced by the metabolic products of microorganisms and which must be important to consider in the cycle.
- In figure 5, lines 1234 and 1235 spp. should'nt be in italics

Reviewer #2 (Comments for the Author):

In this manuscript, Diaz-Gonzalez et al. describe a meta-analysis of known Bacteria/Archaea that are dissimilatory iron reducers, oxidizers, or those that perform both metabolisms. The work represents a significant amount of effort to summarize available literature and data for iron cycling organisms, while also putting forth a trait-based guild framework that could be used to structure the conceptual understanding of microorganisms involved with dissimilatory iron metabolisms. As such, the work is a valuable resource for researchers that are investigating these organisms. The work also represents a novel approach to merge physiological, ecological, and phylogenetic datasets. Consequently, I think this will be a valuable contribution overall. I have some minor comments and suggestions for the authors to consider below that are organized as general and specific comments.

General

The results/discussion about iron preference on lines 306-330 is a bit confusing. My understanding is that these data are based off of media formulations that were used to grow organisms (which were searched and compiled using automated scripts), but the paragraph presents the results as optimal iron preferences. It's entirely possible if not likely that many media formulations just follow recipes that have been previously published, but in my experience, they do not generally identify the optimal concentrations of substrates for organisms. This tends to be pretty rare in characterization studies. In many cases, substrates are provided in significant excess of what the organisms are optimally adapted to. The authors should clarify this and perhaps provide a better caveat if the data is really just the concentration used in the media.

As with all metaanalyses, a significant caveat is that the results are accurate in as much as we already have a comprehensive and accurate view of the study organisms, and I would argue that this may not necessarily be true for any microbial metabolic guild, but perhaps especially iron reducers/oxidizers, because there is comparatively much less known about their physiological strategies in conducting the metabolisms compared to other guilds (e.g., sulfate reducers, nitrate reducers, methanogens, etc). I applaud the author's attempt to quantify guild traits, but I think it's worth a pretty big caveat that additional discoveries/representation of known iron cycling organisms could have the potential to significantly change the correlative results.

In general, there is a fair amount of speculation in the results section that seems better fit for the discussion section. For example the entire paragraph starting on line 409 seems to be discussion material. The authors should consider better delineating what are strictly results vs. interpretation of results (which should be in the discussion section since the results/discussion are separated).

Line 541 - could this be related to the high insolubility of ferric iron at high pH? Iron, perhaps more than any other microbial dissimilatory metabolism substrate, has such high dependence of availability based on environmental conditions (pH/redox), that it would seem that the environments themselves really dictate what is possible for microorganisms to use as substrates. Similar sort of statement could be made about line 590 - ferrous iron doesn't really exist at high pH in thermal environments. I personally think a bit of a stronger emphasis on the role of environments setting the stage for what is possible for microbial metabolism would be useful - but this is of course up to the authors in how they would like to present their framework.

Line 576 and onwards - but this is almost certainly due to cultivation strategy bias and the relative paucity of thermophile researchers, one would imagine?

Line 582 - An argument could be made that this because most researchers don't work with these types of cultures - it's a relatively small field of researchers that study iron cycling in low-temperature environments.

Specific

Line 31 - MIRC guilds should probably be defined here in the abstract for clarity

Line 153 - Typically proprietary refers to private, unshared resources, although this script appears to be publicly available, so perhaps a different term might be more appropriate here

Table 2 - I don't understand what the habitat dominance column is referring to and whether it is necessary. There are clearly lots of other environments where these organisms exist (as expanded upon in the manuscript), so it's unclear to me why these specific environments are highlighted in this table. Authors should evaluate whether it is necessary or adds undue confusion.

Line 311 - This sentence appears to be incomplete - presumably discussing bimodal peaks in the density analyses

Line 371 - perhaps "known" is a better term here than recognizable

line 390 - could an alternative explanation be that higher temperature environments probably generally have more iron because they tend to be hydrothermal, which generally leads to higher iron concentrations in fluids? Thus it may not necessarily be a protein stability issue, but an environmental context issue

Lines 429 & 436 - unclear why these terms are bolded

Reviewer #3 (Comments for the Author):

This study represents a substantial contribution to the topic and represents a valuable potential resource to the research community. I have one major suggestion for addition to the discussion and some minor points.

The major point is to include discussion relating to the mechanism of the nitrate-dependant iron oxidation - to my knowledge, across the literature for this metabolism, especially in the model organisms for NDFO, it has never been definitively shown that the process of iron oxidation is linked to energy generation and is not instead potentially a side-effect of nitrite generation. As this represents a major difference between the NDFO clade and the acidophiles, this gap in our current understanding may be worth highlighting. This is highly relevant to some of the discussion around lines 507 and lines 541.

The minor points are:

Line 72 - state all three states, as you state two.

Line 255 - than for O/R taxa rather than that for O/R taxa

Line 405 - the label (cluster iv) appears to not be used in this section and the labels might benefit from numerical ordering

Response to Reviewers – Manuscript ID mSystems01488-25

Dear Dr. Hernández and Reviewers,

We thank the Editor and Reviewers for their careful evaluation of our manuscript and for their constructive comments. We are encouraged by the consensus that this work represents a valuable contribution to the field of microbial iron cycling. Below, we address each comment point-by-point. All changes are indicated in the revised manuscript (unmarked_manuscript.docx) and highlighted in the marked-up version (marked_up_manuscript.docx). Here, amendments and additions made in response to reviewers' comments are highlighted in red (Reviewer 1), blue (Reviewer 2), or green (Reviewer 3), and are tagged with the corresponding query numbers. Minor grammatical or style edits are unmarked.

We hope the revisions adequately address the concerns raised and that the improved version meets the expectations of mSystems.

With kind regards,

Raquel Quatrini (on behalf of all co-authors)

Reviewer #1: (Comments for the Author):

The ms "Trait-Based Meta-Analysis of Microbial Guilds in the Iron Redox Cycle" by Díaz González et al., is an interesting work concerning a fundamental redox reaction in nature, the Fe cycle. The work deals with an outstanding systematic work using most of the information concerning the subject and the conclusions are very interesting for microbial ecologists. Some suggestions and comments:

R1.1: Although it is not the aim of the work nor the methodologies used, it might be interesting to mention that there is an important level of oxidation of Fe, mainly in anaerobic conditions, produced by the metabolic products of microorganisms and which must be important to consider in the cycle.

Reply: We agree and have added a short discussion acknowledging indirect or abiotic Fe(II) oxidation mediated by microbial metabolic products (e.g., nitrite, sulfide, reactive oxygen species), which may contribute to iron redox cycling under anaerobic conditions. This addition is included in the Discussion section (lines 512-518), where we explicitly note that such processes fall outside the trait-based guild definitions used here but remain relevant to system-level iron cycling.

R1.2: In figure 5, lines 1234 and 1235 spp. should not be in italics

Reply: Corrected (Fig 5 legend).

Reviewer #2 (Comments for the Author):

In this manuscript, Diaz-Gonzalez et al. describe a meta-analysis of known Bacteria/Archaea that are dissimilatory iron reducers, oxidizers, or those that perform both metabolisms. The work represents a significant amount of effort to summarize available literature and data for iron cycling organisms, while also putting forth a trait-based guild framework that could be used to structure the conceptual understanding of microorganisms involved with dissimilatory iron metabolisms. As such, the work is a valuable resource for researchers that are investigating these organisms. The work also represents a novel approach to merge physiological, ecological, and phylogenetic datasets. Consequently, I think this will be a valuable contribution overall. I have some minor comments and suggestions for the authors to consider below that are organized as general and specific comments.

General

R2.1: The results/discussion about iron preference on lines 306-330 is a bit confusing. My understanding is that these data are based off of media formulations that were used to grow organisms (which were searched and compiled using automated scripts), but the paragraph presents the results as optimal iron preferences. It's entirely possible if not likely that many media formulations just follow recipes that have been previously published, but in my experience, they do not generally identify the optimal concentrations of substrates for organisms. This tends to be pretty rare in characterization studies. In many cases, substrates are provided in significant excess of what the organisms are optimally adapted to. The authors should clarify this and perhaps provide a better caveat if the data is really just the concentration used in the media.

Reply: Point taken. We have revised the Methods (lines 149-153) and Results (lines 301-303) to explicitly state that reported iron concentrations correspond to the total Fe supplied in culture media formulations used during physiological characterization, rather than experimentally determined physiological optima, which are rarely assessed explicitly in strain descriptions. We further added a caveat in the Discussion (lines 639-644) acknowledging that media recipes often reflect inherited cultivation practices and may not represent optimal substrate concentrations. Despite this limitation, we retain that the broad dynamic range of reported Fe concentrations and the consistent differences observed among guilds remain informative, as they reflect well-established ecological settings and experimental regimes under which distinct iron redox strategies have been characterized.

R2.2: As with all metaanalyses, a significant caveat is that the results are accurate in as much as we already have a comprehensive and accurate view of the study organisms, and I would argue that this may not necessarily be true for any microbial metabolic guild, but perhaps especially iron reducers/oxidizers, because there is comparatively much less known about their physiological strategies in conducting the metabolisms compared to other guilds (e.g., sulfate reducers, nitrate reducers, methanogens, etc). I applaud the author's attempt to quantify guild traits, but I think it's worth a pretty big caveat that additional discoveries/representation of known iron cycling organisms could have the potential to significantly change the correlative results.

Reply: We fully agree and have strengthened the Discussion (lines 435-444) to emphasize that the proposed guild structure reflects the current state of knowledge and is expected to evolve as additional taxa, traits, and physiological characterizations become available.

R2.3: In general, there is a fair amount of speculation in the results section that seems better fit for the discussion section. For example the entire paragraph starting on line 409 seems to be discussion material. The authors should consider better delineating what are strictly results vs. interpretation of results (which should be in the discussion section since the results/discussion are separated).

Reply: We have revised the Results section to restrict it to descriptive findings and relocated speculative statements to the Discussion section. Limited explanatory phrasing was retained where necessary to aid clarity and continuity of the results.

R2.4: Line 541 - could this be related to the high insolubility of ferric iron at high pH? Iron, perhaps more than any other microbial dissimilatory metabolism substrate, has such high dependence of availability based on environmental conditions (pH/redox), that it would seem that the environments themselves really dictate what is possible for microorganisms to use as substrates. Similar sort of statement could be made about line 590 - ferrous iron doesn't really exist at high pH in thermal environments. I personally think a bit of a stronger emphasis on the role of environments setting the stage for what is possible for microbial metabolism would be useful - but this is of course up to the authors in how they would like to present their framework.

Reply: Most likely. This point is now explicitly addressed in the Discussion (lines 540-543 and 595-598), where we emphasize that environmental physicochemical conditions set the boundary conditions within which microbial iron metabolisms can operate.

R2.5: Line 576 and onwards - but this is almost certainly due to cultivation strategy bias and the relative paucity of thermophile researchers, one would imagine?

R2.6: Line 582 - An argument could be made that this because most researchers don't work with these types of cultures - it's a relatively small field of researchers that study iron cycling in low-temperature environments.

Reply to R2.5 and R2.6: Indeed. We agree with the reviewer in that the apparent scarcity of thermophilic and psychrophilic iron-cycling microorganisms may, at least in part, reflect cultivation strategy biases and the relatively small number of research groups working on iron cycling at temperature extremes. However, because the currently available data do not allow us to quantitatively disentangle true physiological constraints from research and sampling bias, we adopted a conservative interpretation. Accordingly, we retained the phrasing that such taxa are rare, under sampled, or under reported, rather than inferring ecological insignificance. The revisions made in response to comment R2.4 explicitly address these related concerns.

Specific

R2.7: Line 31 - MIRC guilds should probably be defined here in the abstract for clarity

Reply: Done.

R2.8: Line 153 - Typically proprietary refers to private, unshared resources, although this script appears to be publicly available, so perhaps a different term might be more appropriate here

Reply: "Proprietary" has been replaced with "custom" (line 148).

R2.9: Table 2 - I don't understand what the habitat dominance column is referring to and whether it is necessary. There are clearly lots of other environments where these organisms exist (as expanded upon in the manuscript), so it's unclear to me why these specific environments are highlighted in this table. Authors should evaluate whether it is necessary or adds undue confusion.

Reply: The "habitat dominance" column in Table 2 summarizes the most frequently reported environmental contexts in which members of each first-order MIRC guild have been isolated, enriched, or observed, based on the cumulative literature and metadata surveyed. For clarity, the alluded column has been renamed as "Predominant habitat association".

R2.10: Line 311 - This sentence appears to be incomplete - presumably discussing bimodal peaks in the density analyses

Reply: Sentence completed (line 304-306).

R2.11: Line 371 - perhaps "known" is a better term here than recognizable

Reply: "Recognizable" has been replaced with "known" (line 361).

R2.12: Line 390 - could an alternative explanation be that higher temperature environments probably generally have more iron because they tend to be hydrothermal, which generally leads to higher iron concentrations in fluids? Thus it may not necessarily be a protein stability issue, but an environmental context issue

Reply: We agree with the reviewer that environmental context provides a compelling alternative explanation for the observed negative association between siderophore-related traits and temperature. This consideration has been included in the revised manuscript (lines 380-383).

R2.13: Lines 429 & 436 - unclear why these terms are bolded

Reply: Bold formatting removed (lines 412, 419).

Reviewer #3 (Comments for the Author):

This study represents a substantial contribution to the topic and represents a valuable potential resource to the research community. I have one major suggestion for addition to the discussion and some minor points.

R3.1: The major point is to include discussion relating to the mechanism of the nitrate-dependant iron oxidation - to my knowledge, across the literature for this metabolism, especially in the model organisms for NDFO, it has never been definitively shown that the process of iron oxidation is linked to energy generation and is not instead potentially a side-effect of nitrite generation. As this represents a major difference between the NDFO clade and the acidophiles, this gap in our current understanding may be worth highlighting. This is highly relevant to some of the discussion around lines 507 and lines 541.

Reply: We have added a focused paragraph in the Discussion (lines 494-502) highlighting that, for nitrate-dependent Fe(II) oxidation, it remains unresolved whether iron oxidation is directly coupled to energy conservation or occurs indirectly via nitrite-mediated reactions. We contrast this uncertainty with acidophilic iron oxidizers, where Fe(II) oxidation is clearly linked to energy generation.

The minor points are:

R3.2: Line 72 - state all three states, as you state two.

Reply: All three states are now explicitly listed (lines 70-72).

R3.3: Line 255 - than for O/R taxa rather than that for O/R taxa.

Reply: Grammar corrected (line 247).

R3.4: Line 405 - the label (cluster iv) appears to not be used in this section and the labels might benefit from numerical ordering.

Reply: We have revised the text so that clusters are now discussed in numerical order, consistent with their labeling in Figure 4, and have ensured that all cluster labels are explicitly used and clearly referenced in this section (lines 393-401).

Re: mSystems01488-25R1 (**Trait-Based Meta-Analysis of Microbial Guilds in the Iron Redox Cycle**)

Dear Dr. Raquel Quatrini:

Your manuscript has been accepted, and I am forwarding it to the ASM production staff for publication. Your paper will first be checked to make sure all elements meet the technical requirements. ASM staff will contact you if anything needs to be revised before copyediting and production can begin. Otherwise, you will be notified when your proofs are ready to be viewed.

Sincerely,
Marcela Hernandez
Editor
mSystems

Reviewer #4 (Comments for the Author):

Thank you for your detailed revisions to the manuscript and discussion surrounding the NDFO clade.